# Whole Genome Analysis of SLs Pathway Genes and Functional Characterization of *DlSMXL6* in Longan Early Somatic Embryo Development

**DOI:** 10.3390/ijms232214047

**Published:** 2022-11-14

**Authors:** Xueying Zhang, Chunwang Lai, Mengyu Liu, Xiaodong Xue, Shuting Zhang, Yan Chen, Xuechen Xiao, Zihao Zhang, Yukun Chen, Zhongxiong Lai, Yuling Lin

**Affiliations:** Institute of Horticultural Biotechnology, Fujian Agriculture and Forestry University, Fuzhou 350002, China

**Keywords:** *Dimocarpus longan*, SLs, early somatic embryogenesis, *DlSMXL6*

## Abstract

Strigolactones (SLs), a new class of plant hormones, are implicated in the regulation of various biological processes. However, the related family members and functions are not identified in longan (*Dimocarpus longan* Lour.). In this study, 23 genes in the CCD, D27, and SMXL family were identified in the longan genome. The phylogenetic relationships, gene structure, conserved motifs, promoter elements, and transcription factor-binding site predictions were comprehensively analysed. The expression profiles indicated that these genes may play important roles in longan organ development and abiotic stress responses, especially during early somatic embryogenesis (SE). Furthermore, GR24 (synthetic SL analogue) and Tis108 (SL biosynthesis inhibitor) could affect longan early SE by regulating the levels of endogenous IAA (indole-3-acetic acid), JA (jasmonic acid), GA (gibberellin), and ABA (abscisic acid). Overexpression of *SMXL6* resulted in inhibition of longan SE by regulating the synthesis of SLs, carotenoids, and IAA levels. This study establishes a foundation for further investigation of SL genes and provides novel insights into their biological functions.

## 1. Introduction

Strigolactones (SLs) are defined as carotenoid-derived plant hormones, which are divided into oxygen-free carotenes, such as b-carotene, and xanthophylls that contain oxygen, such as lutein [1,2]. SLs were originally identified in root parasitic plants as root communication signals between parasites and symbionts and were rediscovered as phytohormones involved in plant development in recent years [3,4,5,6]. SLs play important roles in plant growth and stress responses, such as seed germination, internode elongation, anthocyanin accumulation, secondary growth, shoot branching, leaf development, and adventitious rooting [7,8,9,10,11,12,13,14,15,16]. β-carotene isomerase DWARF27 (D27), carotenoid cleavage dioxygenases, CCD7 (MAX3, RMS5 D17/HTD1, DAD3), and CCD8 (MAX4, RMS1, D10, DAD1) are involved in the biosynthesis of SLs [17,18,19]. Reportedly, D27-catalysed isomerisation of all-trans-β-carotene is the first step of carlactone formation [20], and D27 was demonstrated to be responsible for SL biosynthesis in plastids [19,21,22]). D27 is a component of the MAX/RMS/D pathway and plays an essential role in biosynthesizing strigolactones, *D27* acts upstream of *MAX1* in the SL biosynthetic pathway in Arabidopsis [22]. The mutation in *D27* leads to a deficiency in strigolactone biosynthesis [23,24]. In rice, the levels of D27 mRNAs are highly elevated in roots under P deficiency and contribute to the high levels of SLs [25]. D27 also plays an important role in adaptation to sulfur deficiency in rice [26]. CCD1, CCD4, and CCD7 have broad substrate specificity, while CCD8 may be specific for SL synthesis to produce several forms of SLs [18,27,28,29,30]. Endogenous SL levels are very low in *ccd7* and *ccd8* mutants [5,6]. Studies demonstrated that mutants *ccd7* and *ccd8* are deficient in SL and exogenous application of GR24, a synthetic analogue to SL, and rescued the excess branching phenotype of *ccd7* and *ccd8* [31].

It was demonstrated that SLs employ SMXL (SUPPRESSOR OF MAX2-LIKE) family members for signal transduction in Arabidopsis [32,33], and SMXL2 plays critical roles in seedling photomorphogenesis and downstream gene expression [34]. SMXL can affect shoot and root architecture and leaf shape as a transcriptional repressor [35,36,37,38,39]. Meanwhile, SMXL6 can function as a transcription factor in SL signalling, which can directly bind to the promoters of *SMXL6*, *7* and *8* in Arabidopsis [40].

Plant somatic embryogenesis (SE) is defined as a general cultivation technique in vitro, where embryos develop from somatic cells without gamete formation and fertilisation [41,42]. Recently, SE became a model system to understand embryonic reproductive programs and epigenetic reprogramming in plants [43]. Plant hormones play important roles in plant SE [44]. Auxin biosynthesis and polar transport are foundational for somatic embryo formation [45,46]. The functional mechanism of SL and auxin biosynthesis, transport, and signalling was extensively studied [47,48]. SL could interfere with auxin polar transport by modifying the PIN-FORMED (PIN) auxin transporter phenotypic output, thereby regulating growth and developmental responses in pea and Arabidopsis [28,49]. In Arabidopsis, an exogenous SL analogue (GR24) inhibits auxin transport, and SLs act downstream of auxin biosynthesis [49]. The results of a previous study demonstrate that SLs can improve the embryogenic process in *Arabidopsis*, as well as GR24 and Tis108 (an SL biosynthetic inhibitor) on SE correlated with changes in the expression of AUXIN RESPONSIVE FACTORs, which are required for the production of embryogenic tissue [50]. These studies suggest that the interaction between SL and auxin plays an important role in plant SE.

The interactions between SLs and other plant hormones were also reported. In tomato, the level of SLs decreased after treatment with specific ABA inhibitors [51]. Conversely, chemical inhibition of SL biosynthetic enzymes CCD7 and CCD8 did not change root ABA contents [51], indicating a complex regulatory mechanism between ABA and SL signalling pathways. Red to far red (R/FR) light sensing positively influences the arbuscular mycorrhizal (AM) symbiosis of a legume and a non-legume through JA and SL signalling [52]. Previous studies showed that gibberellin (GA) plays an important role in the induction of SE [53,54,55]. Reportedly, the GA signalling pathway can regulate the biosynthesis of SLs [56]. However, the interactions between SLs and other plant hormones during early SE are not uncovered, especially in perennial woody plants.

*Dimocarpus longan* Lour is cultivated widely for tropical/subtropical evergreen fruit trees. Since the establishment of the longan SE system by Lai [57,58], which provided ideal experimental materials for studying SE in woody angiosperms, SL genes were found to participate in a wide range of plant developmental processes, including shoot architecture, root growth, photomorphogenesis, leaf senescence, and flower development [11]. However, SL signalling and biosynthesis members and functions are not identified in longan SE. Therefore, based on whole-genome identification and bioinformatics, analysis of the genes involved in the SL pathway was carried out. GR24 (1μM) and Tis108 (3 μM) treatments on longan ECs were carried out (embryogenic callus) to analyse their effects on the transformation of longan ECs into GEs (globular embryos). Microscopic observations showed that GR24 promoted the differentiation of longan ECs, while Tis108 delayed the process, and IAA (indole-3-acetic acid), JA (jasmonic acid), GA (gibberellin), and ABA (abscisic acid) participated in the regulation. Meanwhile, exogenous hormones also regulated the expression levels of SL-related family members and the content of endogenous SLs during longan SE. Expression pattern analysis revealed that these genes were involved in the development of different organs and responded to heat and light. Overexpression of longan *SMXL6* (*DlSMXL6*) inhibited the differentiation of longan somatic embryos, which may be related to the synthesis of IAA by upregulating the expression of longan *YUC5* (*DlYUC5*) and longan *YUC10* (*DlYUC10*). These findings provide a valuable resource for better understanding the biological roles of SL biosynthesis and response genes in longan SE.

## 2. Results

### 2.1. Identification and Chromosomal Localisation Analysis of Genes Involved in the SL Pathway in Longan

In this study, a total of 23 genes were identified in the SL biosynthesis and signalling pathway, including 11 members in the CCD gene family (DlCCD), eight members in the SMXL gene family (DlSMXL), and four members in the D27 gene family (DlD27). To gain insight into the biological functions of DlCCD, DlSMXL, and DlD27 family members, the protein physicochemical properties of the three gene families were further analysed (Appendix A). DlCCD, DlSMXL, and DlD27 encoded proteins with lengths of 526~625 aa, 153~1134 aa, and 142~280 aa, respectively. The MWs of these 23 proteins ranged from 15.4 to 123.80 kDa. The pI ranged from 4.8 to 10.59. The grand average of hydropathicity (GRAVY) values of all members were negative, indicating that they were hydrophilic proteins.

Most family members were predicted to localise in the chloroplast. *DlCCD6* and *DlCCD7* were found in peroxisomes. *DlSMXL6* was predicted to localise in the extracellular matrix. Four members were found primarily in the nucleus, including *DlSMXL2*, *DlSMXL3*, *DlSMXL7*, and *DlD27c*. In addition, no signal peptides exist for all full members of the three gene families, suggesting that none of the DlCCD, DlSMXL, or DlD27 family members were secreted proteins.

As shown in Figure 1A, 23 putative DlCCD, DlSMXL, and DlD27 family members were unevenly distributed on 10 of the 15 chromosomes of longan. The highest number of genes was observed on chromosome 1 (Chr1), including *DlCCD1*, *DlCCD2*, *DlCCD3*, *DlSMXL,1* and *DlSMXL2*. DlSMXL family genes were located on Chr1, Chr5, Chr7, Chr12, and Chr14. The DlD27 genes were located on Chr5, Chr7, and Chr15. Additionally, Chr4, Chr10, Chr12, and Chr15 only contained one number.

Gene duplication is an important pathway for the amplification in family gene numbers and the source of diverse phenotypes [59]. A previous study showed that 7.59% of tandem repeats were identified in the longan genome [60]. Among the 23 genes in the longan genome, only two pairs in the DlSMXL family had a collinearity relationship (Figure 1A), including *DlSMXL3-DlSMXL7* and *DlSMXL4-DlSMXL8*, which were both classified in subgroup VI of the phylogenetic tree (Figure 2). All paralogous homologous genes were derived from segmental duplication, and no tandem duplication events occurred. The Ka/Ks values of the homologous pairs *DlSMXL3-DlSMXL7* and *DlSMXL4-DlSMXL8* were 0.24 and 0.16, respectively, suggesting that these two gene pairs were subject to strong purifying selection and were functionally stable. Furthermore, colinear analysis of longan and Arabidopsis genomes suggests that 13 pairs of collinear genes exist between longan and Arabidopsis. Among them, *DlCCD8* (Chr10) and *DlSMXL7* (Chr12) corresponded to two genes in Arabidopsis (Figure 1B).

### 2.2. Phylogenetic Analysis and Classification of CCD, SMXL, and D27 Gene Families

To gain insight into the evolutionary relationships and predict the functions of the plant genes involved in SL biosynthesis and signalling pathways, we next compared the full length of these protein sequences as found in *Dimocarpus longan* Lour, Arabidopsis, *Oryza sativa*, *Citrus sinensis*, *Zea mays*, *Glycine max*, *Solanum lycopersicum*, and *Triticum aestivum*. We constructed phylogenetic trees (Figure 2). Based on the amino acid sequence similarity and topological structure of the phylogenetic tree with MEGA 8 software, these genes were divided into six subfamilies, including Group I to Group Ⅵ. In our study, phylogenetic analysis showed that Group I only belonged to DlCCD5, which may be due to the unique genetic structure. SMLX family genes were highly conserved, as well as homologous in longan and Arabidopsis, which mostly belonged to Group III. DlCCDs were distributed in Group I~Group VI, indicating diversity during evolution.

### 2.3. Gene Structure and Motif Composition of the CCD, SMXL, and D27 Gene Families in Longan

Gene structural diversity and conserved motif divergence are possible mechanisms for the evolution of multigene families [61]. To further explore the diversity and functional differentiation of the DlCCD, DlSMXL, and DlD27 gene family members, gene structure and protein-conserved motif analyses were performed (Appendix A). Gene structure analysis showed that DlD27 gene family members were scattered both in exons and introns, and the first intron fell within the mature coding sequence, which facilitated the neat separation of the signal sequence from the coding sequence. *DlCCD1*, *DlCCD7*, *DlCCD9*, *DlSMXL1*, *DlSMXL2*, *DlSMXL4*, and *DlSMXL7* contained the 5′-UTR, CDS and 3′-UTR, respectively, with a good gene structure. However, none of the DlD27 family members had a 5′-UTR or 3′-UTR, which may affect gene transcription and translation. Analysis of the conserved motif showed that *DlCCD* family members contained 7 or 10 motifs, and motif 1, motif 2, motif 7, motif 8, motif 9, and motif 10 were present in all CCD family members, indicating a high conservation in longan evolution. *DlSMXL5* and *DlSMXL6* only contained motif 15 and motif 12, respectively. DlD27 family genes all only contained one motif, and *DlD27a*, *DlD27c*, and *DlD27d* all contained motif 16.

### 2.4. Cis-Acting Elements in Longan CCD, DlSMXL, and D27 Family Genes in Longan

To further analyse the potential function of the DlCCD, DlSMXL, and DlD27 genes, a 2000-bp regulatory region upstream of the ATG (promoter) was searched for *cis*-acting element analysis (Figure 3). Among the promoter *cis*-acting elements of the three gene family members, the largest number of photoresponse elements were present in all members of the three gene families. In addition, responsive cell cycle elements were found only in *DlCCD3*, which may play an important role in longan cell cycle progression. Most of the members responded to ABA, MeJA, GA, SA, and auxin. Three gene family members also had elements in response to low temperature, drought, stress, seed-specific regulation, endosperm expression, and meristem expression, indicating that these genes may play important roles in reverse resistance, cold tolerance, seed growth, and embryonic mature development.

The *cis*-acting elements of the promoter could combine with specific transcription factors and then form a transcription initiation complex to initiate gene-specific expression in various biological processes [62]. To investigate the regulation of transcription factors (TFs) on the expression of DlCCD, Dl27, and DlSMXL family genes, transcription factor-binding sites (TFBSs) on the promoter were predicted using the PlantTFDB online tool. In total, 44, 42, and 42 TF families were identified in the DlCCD, DlD27, and DlSMXL family gene promoters, respectively. With the exception of GRF and LFY on DlCCD family gene promoters, most TFBSs were identified on DlCCD, DlD27, and DlSMXL family gene promoters, such as AP2, ARF, BBR-BPC, bZIP, bHLH, Dof, ERF, NAC, WOX, WRKY, MYB, and TCP; this indicates a potential common upstream regulatory mechanism of these genes in longan growth and development and various stress responses, especially in regulating longan early somatic embryogenesis.

### 2.5. RNA-Seq Revealed the Expression Profiling of Longan CCD, SMXL, and D27 Family Genes in Different Tissues and Treatments

We further analysed the expression patterns of DlCCD, DlSMXL, and DlD27 family genes in different tissues and treatments by transcriptome data (Figure 4). The results show that most genes were highly expressed in stems (Figure 4A), which might be related to the widespread involvement of SL in plant shoot branching. *DlCCD2*, *DlCCD3*, *DlCCD11*, and *DlSMXL2* were highly expressed in the flower. *DlCCD2*, *DlCCD4*, and *DlSMXL7* were highly expressed in the leaf. *DlD27a*, *DlSMXL6*, and *DlSMXL7* were highly expressed in the roots. *DlCCD1*, *DlCCD3*, *DlSMXL1*, *DlSMXL2*, *DlSMXL6, DlSMXL7*, *DlD27a*, and *DlD27d* were highly expressed in the seed and might be involved in longan seed dormancy and germination. *DlSMXL2*, *DlCCD5*, *DlCCD6*, and *DlCCD7* were highly expressed in the young fruit, indicating that these genes might be involved in fruit development and maturation.

A large number of photoresponsive elements were identified in the three gene family member promoters (Figure 3), and transcriptome data show that *DlCCD2*, *DlCCD6*, *DlCCD7*, *DlCCD11*, *DlSMXL3*, and *DlSMXL6* were highly expressed under blue light. *DlD27a*, *DlD27d*, *DlCCD5*, *DlCCD8*, *DlCCD10*, *DlSMXL5*, and *DlSMXL7* were highly expressed under white light (Figure 4B). In addition, the three gene family members showed different expression patterns under different temperature treatments. Moreover, with the exception of *DlSMXL2*, *DlSMXL3*, *DlCCD5*, and *DlCCD8*, all three gene family members were upregulated in response to high-temperature stress (35 °C), indicating that these genes might be involved in cell self-repair under high temperature (Figure 4C).

### 2.6. Genome Expression Profiling of Longan CCD, SMXL and D27 Family Genes and SL Content Determination during Early SE in LONGAN

To further investigate the potential functions of members of the DlCCD, DlSMXL, and DlD27 gene families in longan SE, a heatmap was drawn according to the FPKM values of the genes from the longan transcriptome database (Figure 5B). The results show that *DlCCD3*, *DlCCD5, DlCCD7*, *DlCCD9*, and *DlCCD10* were highly expressed in the ECs stage, while *DlCCD6* was highly expressed in both the ECs and ICpEC stages. The expression of *DlCCD11* was significantly higher in the ICpEC stage than in the EC sand GEs stages, while *DlCCD1*, *DlCCD2*, *DlCCD4*, and *DlCCD8* were highly expressed in the GE stage. In the DlSMXL gene family, *DlSMXL4* and *DlSMXL8* were both expressed at low levels in the three stages. *DlSMXL7* was highly expressed in the ECs stage, *DlSMXL5* and *DlSMXL6* were highly expressed in the ICpEC stage, and *DlSMXL1*, *DlSMXL2*, and *DlSMXL3* were highly expressed in the GEs stage. *DlD27b* and *DlD27c* were expressed at low levels in all three stages of longan somatic embryo development, while *DlD27a* was highly expressed in the GE stage, and *DlD27d* was expressed at high levels in both ECs and GEs. The qRT–PCR results show that the genes had the same expression trends as those obtained by RNA-seq (Figure 5C). The SL content increased significantly during longan SE, with the highest content in GEs, approximately 1.26 times that in ECs (Figure 5D). Taken together, the results indicate that the SL pathway may be involved in longan early SE.

### 2.7. Effects of Exogenous Hormones on CCD, SMXL and D27 Family Gene Expression and SL Content in Longan SE

According to the *cis*-acting element analysis of the DlCCD, DlSMXL and DlD27 families, we chose 10 validated genes by qRT–PCR to confirm whether the expression patterns were influenced by hormone treatments (IAA, ABA, GA, and JA) for 24 h. We first determined the endogenous SL content of longan ECs under different hormone treatments. The results show that exogenous IAA and GA significantly increased the SL content compared to the control, while ABA and JA decreased the SL content in longan ECs (Figure 6B).

Furthermore, we found that the expression patterns of DlCCD, DlSMXL and DlD27 family genes under different hormone treatments differed (Figure 6A, Appendix A). The results show that the expression of *DlCCD4*, *DlCCD8* and *DlD27a* was upregulated, while *DlSMXL2* and *DlSMXL6* were downregulated by IAA treatment. *DlCCD5*, *DlCCD8* and *DlD27d* were upregulated, and *DlSMXL2* was downregulated by GA treatment. ABA inhibited the expression of *DlCCD5*, *DlCCD7*, and *DlD27d*, and treatment downregulated the expression of *D27a* and upregulated the expression of *DlSMXL6*. These results indicate that these genes could respond to hormones and participate in SL biosynthesis and the accumulation of longan ECs.

### 2.8. Effects of Exogenous GR24 and Tis108 on Longan SE

To further investigate the function of SL in longan early SE, the synthetic strigolactone analogue GR24 (1 μM) and SL biosynthetic inhibitor Tis108 (3 μM) were applied in this study (Figure 7). The microscopic examination results show that EC cells were already at the ICpEC stage under GR24 treatment, while the EC cells were still loose under Tis108 treatment at 6 days. A few globular embryos were already appeared at 8 d under the GR24 treatment. The typical GE structure appeared in both the control group and GR24 treatment, however, the more numbers and higher compactness of the GEs were observed under GR24 treatment, while materials had not reached the GEs stage under Tis108 treatment at 10 days (Figure 7A).

To further explore the interaction relationship between SLs and other hormones in longan, endogenous hormone contents were determined with the SL synthesis analogue GR24 and the synthesis inhibitor Tis108. The results show that GR24 decreased the IAA levels of longan somatic embryos compared with the control, while Tis108 increased the IAA levels at 6 and 8 days (Figure 7C). These results further suggest that SLs played important roles in longan early SE, which was accompanied by regulating the synthesis of IAA. Further results show that the content of endogenous ABA was globally decreased upon Tis108 treatment at 8 and 10 days (Figure 7D). The content of GA was significantly decreased at 6 and 10 days upon GR24 and Tis108 treatment (Figure 7D). GR24 significantly increased the JA levels at 6 and 8 days, while Tis108 significantly decreased the JA levels at 10 days (Figure 7E). These results show that GR24 and Tis108 may regulate longan early SE by changing the content of endogenous hormones.

### 2.9. Overexpression of SMXL6 Inhibited Differentiation of the Early Somatic Embryogenesis in Longan

The above results show that *DlSMXL6* expression decreased gradually during longan SE and was downregulated by IAA treatment, indicating that it might play a role in somatic embryo development. To verify this prediction, the full-length *DlSMXL6* CDS was cloned into the pCAMBIA-1301-35S overexpression vector and stably transformed into longan ECs. We selected the OE-1, OE-2, and OE-3 cell lines with high expression levels for further analyses (Figure 8A). Phenotypic observation results show that the *SMXL6-*OE lines could not differentiate into GEs at 12 days; however, the WT and *1301-GUS* lines exhibited typical GEs, revealing that overexpression of *DlSMXL6* inhibited the differentiation of longan somatic embryos (Figure 8B–F). Compared with the WT and *1301-GUS* cell lines, the contents of SLs were significantly decreased (Figure 8H). We further found that the precursors for SL biosynthesis and carotenoids were significantly increased in the *DlSMXL6*-overexpressing cell lines, which may be due to negative feedback regulation (Figure 8G). Furthermore, the levels of IAA were significantly increased in the *DlSMXL6*-overexpressing cell lines (Figure 8I). To further investigate the regulatory mechanism of IAA synthesis in *DlSMXL6*-overexpressing cell lines, we also analysed the expression of IAA biosynthetic genes, including *DlYUC3*, *DlYUC5*, and *DlYUC10*, which were differentially expressed during longan early SE stages based on transcriptome data. Compared with WT and *1301-GUS*, *DlYUC5* and *DlYUC10* were obviously upregulated in the *DlSMXL6*-overexpressing cell lines, which may play important roles in the interaction of auxin and SLs during longan early SE (Figure 8J–L).

## 3. Discussion

### 3.1. CCD, D27, and SMXL Members Are Involved in Early SE and Organ Development in Longan

Strigolactones play crucial roles in signalling transportation and gene expression during plant growth processes [7,63]. CCD, D27, and SMXL family genes are involved in SL signalling and biosynthesis [64,65,66,67,68,69,70]. In Arabidopsis, two SL biosynthetic genes, the *CCD7* and *CCD8* homologous genes *MAX3* and *MAX4*, are upregulated during the induction phase of embryogenesis [50]. In this study, following early SE, SLs accumulated progressively. *DlCCD6* was highly expressed in both ECs sand ICpEC stages, and *DlCCD1*, *DlCCD2*, *DlCCD4*, *DlCCD8*, and *D27a* were highly expressed in the GE stage, which may play important roles in the accumulation of SLs. The repressors of SLs signalling and biosynthesis, the repressors of SL signalling and biosynthesis, *DlSMXL5, DlSMXL6*, and *DlSMXL7*, were expressed at low levels at the GEs stage. These genes may play key roles in regulating SL levels during longan SE (Figure 5).

SLs were found to participate in a wide range of plant developmental processes, including root growth, leaf senescence, photomorphogenesis, and flower development [10,11]. In this study, DlCCD, DlD27, and DlSMXL members showed divergent expression patterns across different tissues of longan. Strigolactone application increased the number of flowers and also promoted flower and fruit size of tomato [14,71]. *MAX1* homologues may be involved in the regulation of flower development [72]. *CCD7* were highly expressed in the tomato fruit [73]. In this study, *DlCCD2*, *DlCCD3*, *DlCCD11*, and *DlSMXL2* were highly expressed in the flower, and *DlSMXL2*, *DlCCD5*, *DlCCD6*, and *DlCCD7* were highly expressed in the young fruit, which might be involved in longan floral induction and fruit development. Strigolactones can induce secondary growth in stem and root tissue [65]. In this study, most genes were highly expressed in stems, and *DlD27a* and *DlSMXL7* were highly expressed in roots, which may participate in the regulation of longan root formation and elongation [74]. Studies show that SLs are involved in the regulation of leaf development and senescence [15,75]. In this study, *DlCCD2*, *DlCCD4*, and *DlSMXL7* were highly expressed in the leaf, which might play a special role in longan leaf development. SL-deficient *ΔCCD7* exudates reduced seed germination activity [76]. In this study, *DlCCD1*, *DlCCD3*, *DlSMXL1*, *DlSMXL2*, *DlSMXL7*, *DlD27a*, and *DlD27d* were highly expressed in the seed, which might be involved in seed dormancy and germination [77,78]. These results provide a foundation for further investigation of DlCCD, DlD27, and DlSMXL members in longan development and maturation.

### 3.2. CCD, D27, and SMXL Genes Respond to Abiotic Stress in Longan Somatic Embryos

The promoter regions of the DlCCD, DlD27, and DlSMXL genes contained a large number of stress-related *cis*-acting elements, indicating that these genes may be widely involved in longan somatic embryo stress responses (Figure 3). The analysis of transcriptome data under high temperature and blue light treatment revealed that the expression patterns of DlCCD, DlD27, and DlSMXL genes under different stresses differed (Figure 4). Light is an important factor that affects the synthesis of functional metabolites in longan ECs [79]. In Arabidopsis, light and strigolactone signalling together contribute to the regulation of branching [80]. In this study, *DlCCD2*, *DlCCD6*, *DlCCD7*, *DlCCD11*, *DlSMXL3*, and *DlSMXL6* were induced by blue light. *DlD27a*, *DlD27d*, *DlCCD5*, *DlCCD8*, *DlCCD10*, *DlSMXL5*, and *DlSMXL7* were induced by white light. Reportedly, strigolactone biosynthesis and signalling play crucial roles in the responses to heat stress in tomato, and high growth temperature significantly induced the transcription of *CCD7* and *CCD8* in the roots at 3 h after heat stress [81]. In this study, most members were upregulated in response to high-temperature stress (35 °C) at 24 h, indicating that these genes might play important roles in cell self-repair under high temperature.

### 3.3. SLs Are Involved in the Regulation of Longan Early SE by Interacting with Endogenous Hormones

Hormones play crucial roles in plant growth and development, especially in plant SE. Recently, the interactions between SLs and other hormones were uncovered in plant development [82,83]. Auxin plays a crucial role in plant SE [45,46,84,85,86]. In most cases, the transcription levels of *CCD7* and *CCD8* are induced by auxin [87,88,89,90]. Upregulation of SL biosynthetic genes (*CCD7*, *CCD8*, and *D27*) by auxin was reported [88,89,90]. In this study, we found that IAA significantly increased the SL content compared to the control by regulating the expression of *DlCCD4*, *DlCCD8*, *DlD27a*, *DlSMXL2*, and *DlSMXL6*, indicating that these genes may be involved in SL synthesis under IAA treatment to regulate longan SE. To further demonstrate the effects of SLs on ECs differentiation and the endogenous hormone content changes during longan SE, GR24 and Tis108 were applied to treat longan ECs. The results show that GR24 could promote the transformation of ECs into GEs, and Tis108 inhibited the process. In this study, GR24 decreased the IAA level of longan embryos, while Tis108 could increase the IAA level, indicating that GR24 and Tis108 could regulate the SE process by altering the endogenous IAA content [49,91].

ABA plays an important role in plant SE, which can accelerate the synthesis and accumulation of nutrients [92,93,94,95,96]. Both ABA and SLs originate from a carotenoid precursor, which indicated that their biosynthesis pathways might be interregulated [97]. Studies show that SL signalling can regulate ABA metabolism and sensitivity [98,99,100]. In this study, *DlCCD5*, *DlCCD7*, and *DlD27d* were downregulated in longan ECs by ABA treatment. GR24 did not lead to an increased content of ABA [99], and negative regulation of ABA metabolism by SLs was reported [97]. In cherry, GR24 can significantly increase the ABA level of roots, whereas Tis108 would decrease the ABA level [100]. However, in this study, the endogenous ABA levels were significantly decreased upon Tis108 treatment at 8 and 10 days, but did not respond to GR24.

GA was reported to be necessary for the induction of SE in spinach [101,102,103,104] The physiological interactions between SL and GA were reported previously [78,105,106,107]. In this study, exogenous GA significantly increased the SL content of longan ECs by regulating the expression of *DlCCD5*, *DlCCD8, DlD27d*, and *DlSMXL2.* GR24 could significantly increase the GA content in seeds [78], and SL can promote the production of bioactive GA [107]. In *Jatropha curcas*, SL acted antagonistically to gibberellin in the control of lateral bud outgrowth [105,106]. However, in this study, both GR24 and Tis108 significantly decreased the GA levels at 6 and 10 days, indicating that GA may not respond to GR24 and Tis108 during longan SE.

It was reported that JA inhibits embryo germination in angiosperms [108,109], and a dual mode of GR24 is found in regulating JA accumulation in rice roots [110]. In this study, exogenous JA significantly decreased the SL content of longan ECs by upregulating the expression of *DlSMXL6* and downregulating the expression of *DlD27a.* The JA content significantly increased upon GR24 treatment at 6 and 8 days and significantly decreased upon Tis108 treatment, which may play an important role in longan SE upon SL regulation.

Taken together, GR24 promoted longan SE by decreasing the levels of IAA and increasing the levels of JA at the early stage, while Tis108 inhibited the process by regulating the levels of IAA at the early stage and ABA and JA at the late stage (Figure 9A). However, the regulatory mechanisms among SL and endogenous hormones in longan SE remain to be further investigated.

### 3.4. Function of SMXL6 in Longan Early Somatic Embryo Development

As a downstream regulatory factor of SL, SMXL6, 7, and 8 are important switches of the SL signalling pathway and play an important role in regulating the growth and development of plant branches, root architecture, and leaf shape [35,36,38,39]. Meanwhile, SMXL6 can function as a transcription factor in SL signalling, which can directly bind to the promoters of *SMXL6*, *7*, and *8* in Arabidopsis [40]. Compared to the progress in other species, little is known about the functions of SL in longan SE. In the present study, *DlSMXL6* was specifically highly expressed in ECs and IcpEC, and expressed at low levels in GE, indicating its role in somatic embryo development. The results show that *SMXL6-*OE cells could not differentiate into GE cells on the 12th day, while WT and *1301-GUS* cells exhibited typical GEs.

It was shown that auxin biosynthesis and polar transport are foundational for somatic embryo formation [45,46]. Auxin is involved in the regulation of SLs, which may play a direct role in meristematic processes [68]. SLs can regulate shoot branching and root formation by changing the auxin optimum needed [24,111], and auxin signalling acts downstream of SLs [13,112]. In this study, the content of IAA was significantly increased in the *DlSMXL6*-overexpressing cell lines compared to the WT and *1301-GUS* cell lines by upregulating the expression of *DlYUC5* and *DlYUC10*. As reported, the requirement of IAA content decreased during the plant SE [113]. Overexpression of *DlSMXL6* in longan ECs also promoted the accumulation of SL precursor carotenoids, which may be due to the feedback regulation of the SL pathway (Figure 9B).

In summary, we first identified SL pathway genes and revealed the expression patterns of different organ development and abiotic stress responses, especially in early SE in longan. We found that GR24 and Tis108 could affect longan early SE by regulating the levels of endogenous hormones. Functional analysis of *DlSMXL6* in longan ECs showed that it was involved in the regulation of SE. However, the regulatory mechanism of *DlSMXL6* in longan SE remains to be further investigated.

## 4. Materials and Methods

### 4.1. Plant Materials

The ‘Hong He Zi’ (‘HHZ’) longan ECs were obtained as previously described [58]. The experiment was conducted at the Institute of Horticultural Biotechnology at Fujian Agriculture and Forestry University from 2020 to 2021. The ECs were cultured in Murashige and Skoog (MS) liquid medium (20 g∙L^−1^ sucrose and 7 g∙L^−1^ agar, pH 5.8~6.0) supplemented with 100 μM IAA, ABA, JA, or GA for 24 h. Additionally, to investigate the effects of SL on the conversion of ECs into GEs in longan, the ECs were cultured in MS liquid medium added with GR24 (1 μM stock in acetone) and Tis108 (3 μM stock in DMSO). The same volume of acetone (Mock1) and DMSO (Mock2) were as controls at 25 °C with 120 r·min^−1^ shaking in the dark for 10 days. The test materials were immediately frozen in liquid nitrogen and stored at −80 °C for subsequent tests. Each assay was repeated three times.

### 4.2. Identification of Genes Involved in SL Biosynthesis and Signalling in Longan

The genes involved in SL biosynthesis and signalling in Arabidopsis were obtained from TAIR (http://www.arabido-psis.org (accessed on 10 May 2021). They were used as base alignment sequences with the local BLAST of TBtools [114]. The ‘HHZ’ longan genome database from the NCBI Sequence Read Archive (SRA) database (SRR17675476) was used for homologyously alignment. All of the sequences identified for SL biosynthesis and signalling genes in longan were subjected to NCBI’s Batch CD-search (https://www.ncbi.nlm.nih.gov/Structure/bwrpsb/bwrpsb.cgi (accessed on 10 May 2021) and SMART (http://smart.embl-heidelberg.de/ (accessed on 15 May 2021)) tools to verify their reliability as targets [115]. We also examined the physicochemical properties of SL biosynthesis and signalling proteins, including determining the protein length (aa), molecular weight (MW), isoelectric point (pI), and GRAVY using ExPASy (https://www.expasy.org/ (accessed on 15 May 2021). The subcellular localisation of SL biosynthesis and signalling proteins were predicted with Plant-PLoc (http://www.csbio.sjtu.edu.cn/bioinf/plant/ (accessed on 15 May 2021) [116]. The signal peptides were analysed with SignalP 4.0 (http://www.cbs.dtu.dk/services/SignalP/ (accessed on 15 May 2021). Chromosomal localisation information was extracted from the GFF file, and the results were visualised with Tbtools. The *cis*-acting elements were analysed with PlantCARE (http://bioinformatics.psb.ugent.be/webtools/plantcare/html/ (accessed on 15 May 2021), which were present from 2000 bp upstream of the gene initiation codon. PlantTFDB (http://planttfdb.cbi.pku.edu.cn/ (accessed on 15 May 2021) was used to predict the transcription factor-binding sites (TFBSs) on promoters with the parameter set of *p* value ≤ 1 × 10^−6^. MEME (https://meme-suite.org/meme/tools/meme (accessed on 15 May 2021) was used to analyse the proteins’ conserved motifs. Finally, TBtools software (v1.098669, South China Agricultural University, Guangzhou, China) was used to annotate the gene structure and visualise the conserved domains and motifs. MCScanX software was used to predict the collinearity of the genes of SL biosynthesis and signalling and to mark the collinear genes [117]. KaKs_Calclator 2.0 software (https://sourceforge.net/ (accessed on 20 May 2021) was used to estimate synonymous (Ks) and nonsynonymous (Ka) substitution rates [118].

### 4.3. Phylogenetic Analysis of the Gene Families Involved in the SL Pathway in Longan

To compare the phylogenetic relationship of the genes involved in SL biosynthesis and signalling in longan with their counterparts in other plant species, we obtained the amino acid sequences of Arabidopsis (*Arabidopsis thaliana*) from TAIR (http://arabidopsis.org (accessed on 20 May 2021) and rice (*Oryza sativa*) from hytozome v12.1 (https://phytozome.jgi.doe.gov/pz/portal.html (accessed on 20 May 2021). The amino acid sequences of *Zea mays*, *Citrus sinensis*, *Glycine max*, *Solanum lycopersicum*, and *Triticum aestivum* were downloaded from NCBI (https://www.ncbi.nlm.nih.gov/ (accessed on 20 May 2021). The phylogenetic trees were constructed using MEGA 5.0 software with neighbour-joining (NJ) [119].

### 4.4. Analysis of the FPKM Expression Profile of Genes Involved in the SL Pathway in Longan

The FPKM expression profile of genes involved in the SL pathway in longan was extracted from four transcriptome datasets, including three for the ‘Hong He Zi’ (HHZ) cultivar and one for the ‘Si Ji Mi’ (SJM) cultivar. RNA-seq data during early longan SE were available in the NCBI SRA repository under accession code SRR21979789, SRR21979788 and SRR21979787 (ECs), SRR21979786, SRR21979785 and SRR21979784 (ICpECs, Incomplete compact proembryogenic culture) and SRR21979783, and SRR21979782 and SRR21979781 (GEs). RNA-seq data of different temperature treatments on ECs are available in the NCBI SRA repository under accession code SRR21921625, SRR21921624 and SRR21921623 (15 °C), SRR21921622, SRR21921621 and SRR21921620 (25 °C), SRR21921619, and SRR21921618 and SRR21921617 (35 °C). The different light qualities (blue-light, whitelight, and dark as the control) on ECs are available in dataset [79]. The ‘SJM’ transcriptome dataset was the six-organ (root, stem, leaf, flower, young fruit, and seed) dataset [60]. The fragments per kilobase of exon model per million mapped fragment (FPKM) values of isoform transcripts were analysed with Excel and visualised with TBtool software [114].

### 4.5. Determination of Endogenous Hormone, Carotenoid, and SL Contents in Longan

The levels of IAA, GA, JA, ABA, carotenoid, and SL were determined with a double antibody one-step sandwich enzyme-linked immunosorbent assay (ELISA). The specific steps were as follows: the standard, sample extract, and horseradish peroxidase (HRP)-labelled detection antibody were added to the coating micropores, their corresponding antibodies were added, and the samples were incubated and washed thoroughly. The substrate 3,3′,5,5′-tetramethylbenzidine was used for colour development. TMB was transformed into blue under the catalysis of peroxidase and finally into yellow under the action of acid. The colour depth was positively correlated with the content of the substance to be measured in the sample. The absorbance was determined with a microplate reader (Infinite M200 PRO, Tecan, Männedorf, Switzerland) at a 450 nm wavelength to calculate the sample activity. Each assay was repeated three times.

### 4.6. RNA Extraction and Real-Time Fluorescence Quantitative Polymerase Chain Reaction (qRT–PCR)

Total RNA was extracted using the TransZol Up reagent kit (TRANS, Beijing, China) and treated with RNase-free DNase I (TRANS, Beijing, China) following the manufacturer’s instructions. The RNA quality was analysed by agarose gel electrophoresis and quantified using a Nanodrop 2000 spectrophotometer (Thermo Scientific, Wilmington, DE, USA). Only RNAs that met the criteria 260/280 ratio of 1.8–2.1 and 260/230 ratio ≥ 2.0 were used for further analyses and stored at −80 °C.

The RNA was used to synthesise first-strand cDNA using a PrimeScript RT Master Mix (Perfect Real Time) cDNA Synthesis Kit (TaKaRa, Japan). DNAMAN software was used to design the qRT–PCR primers (Appendix A). The efficiency of the qPCR was verified for each primer pair using the cDNA of longan somatic embryos (Appendix A). Additionally, qRT-PCRs were performed on a Roche Lightcyler 480 instrument using SYBR Green chemistry (Hieff qPCR SYBR Green Master Mix, YEASEN, China). *DlFeSOD* (EU330204) was used as a reference gene for the internal control [120]. The operating parameters of the qRT–PCR were as follows: 95 °C for 30 s, followed by 40 cycles of 95 °C for 10 s and 58 °C for 30 s. The relative expression of the genes involved in SL biosynthesis and signalling in longan was calculated via the 2^−ΔΔCt^ method [121].

### 4.7. Vector Construction

The *SMXL6* coding sequences in longan ECs were amplified by PCR and then inserted into a separate pCAMBIA-1301-35S vector at the BamH I and Sal I restriction sites using the In-Fusion HD Cloning Kit (Takara, CA, USA). The generated pCAMBIA1301-35S-*SMXL6* recombinant plasmids were used for longan ECs stable transformation.

### 4.8. Stable Transformation of Longan Embryogenic Calli

The pCAMBIA1301-35S-*SMXL6* constructs were transformed into the *Agrobacterium tumefaciens* strain (GV3101) following the manufacturer’s instructions (TRANS, Beijing, China). The rapidly proliferating longan ECs were transferred into the bacterial liquid of *Agrobacterium tumefaciens* (OD600 = 0.7–0.8). After 30 min of infection, the infected ECs were transferred into coculture MS solid medium containing 30 g/L sucrose for three days and then transferred into MS solid medium containing 1.0 mg/L 2,4-D, 20 g/L sucrose and 300 mg/L cefotaxime and harvested after 20 days. The proliferating ECs were transferred into MS solid selection medium containing 1.0 mg/L 2,4-D, 20 g/L sucrose, 300 mg/L cefotaxime, and 20 mg/L hygromycin and subcultured every one month until resistant ECs formed. GUS staining (Huayueyang Biotechnology, Beijing, China) and PCR amplification were used to detect the transformed ECs. For the differentiation of early somatic embryos, longan wild-type and transformed ECs were cultured in MS solid medium containing 300 mg/L cefotaxime, and samples were harvested after 6 and 12 days. The morphological characteristics of transformed somatic embryos were observed by microscopy (Olympus).

### 4.9. Statistical Analyses

Data were analysed, and graphs were drawn using GraphPad Prism 8 (GraphPad Prism Software, Inc., San Diego, CA, USA). In all graphs, error bars indicate standard deviation, and significant differences are indicated with “*” (*p* < 0.05), “**” (*p* < 0.01), “***” (*p* < 0.005), or “****” (*p* < 0.001) using Student’s *t* test.

## Figures and Tables

**Figure 1 ijms-23-14047-f001:**
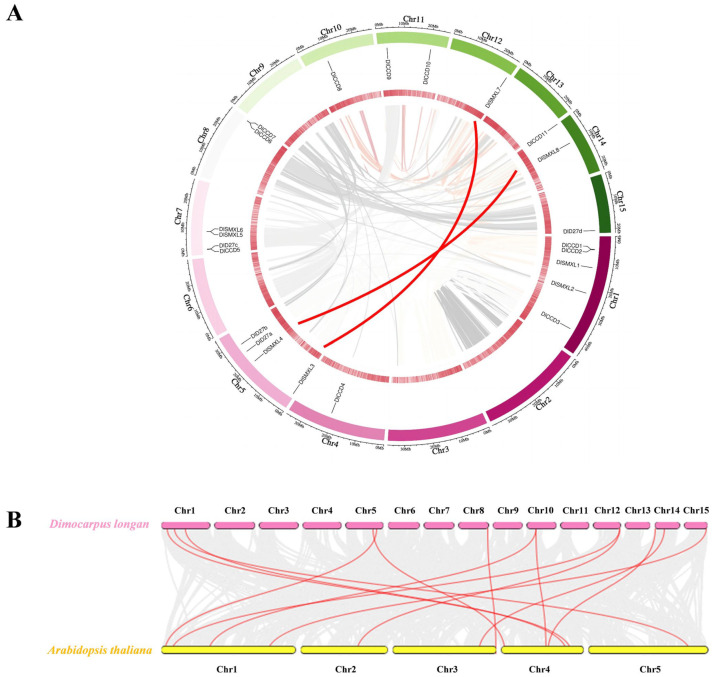
Collinear analysis of DlCCD, DlD27, and DlSMXL family members. (**A**) Collinearity analysis within longan. Gray lines indicate the collinear blocks within the longan genome, the red line represent the syntenic gene pairs relating to DlCCD, DlD27, and DlSMXL family genes. (**B**) Colinear analysis between longan and Arabidopsis. Gray lines indicate the collinear blocks within the longan and Arabidopsis, and the red lines represent the syntenic gene pairs relating to SL biosynthesis.

**Figure 2 ijms-23-14047-f002:**
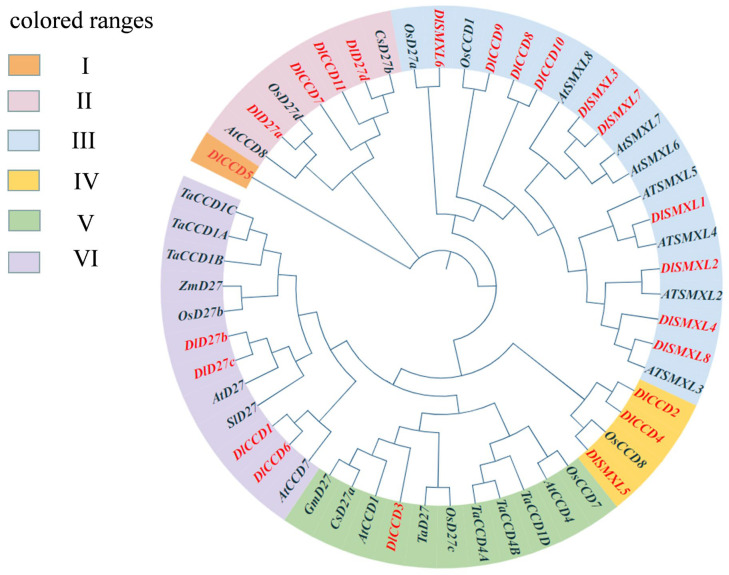
Phylogenetic tree of CCD, SMXL, and D27 family genes. The different-colored areas indicate different groups.

**Figure 3 ijms-23-14047-f003:**
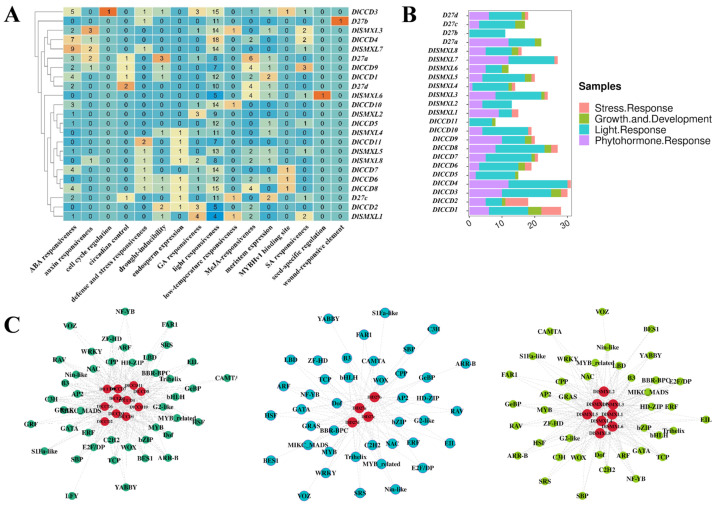
CRE analysis of the promoter region of DlCCD, DlSMXL, and DlD27 genes. (**A**) DlCCD, DlSMXL, and DlD27 family members were arranged in the order of phylogenetic tree. The names of CREs were marked on the bottom. The heatmap shows the number of main CREs corresponding to each member. (**B**) CREs were classified into four categories: stress response, growth and development, light response, and phytohormone response. The bar chart counts all the number of four types of CREs in each member, and different types of CREs are indicated by different colors. (**C**) Network diagram of predicted transcription factor binding sites with DlCCD, DlSMXL, and DlD27 family members. Gray connecting lines indicate the presence of interactions.

**Figure 4 ijms-23-14047-f004:**
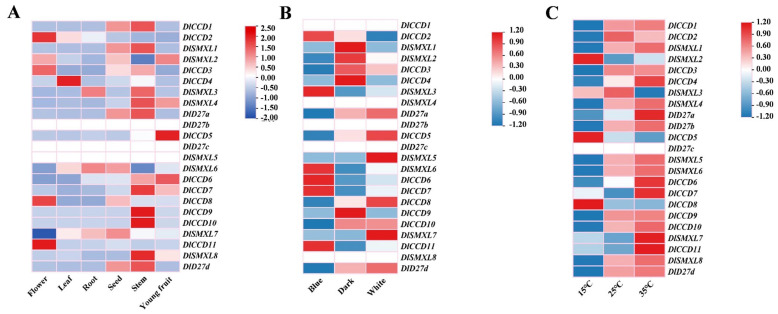
The expression profiling of DlCCD, DlSMXL, and DlD27 genes detected in different tissues and in response to different treatments. (**A**) Transcriptome data of DlCCD, DlSMXL, and DlD27 genes of six organs in longan. (**B**) Transcriptome data of DlCCD, DlSMXL, and DlD27 genes in the ECs under different light quality treatments for 25 d. (**C**) Transcriptome data of DlCCD, DlSMXL, and DlD27 genes in the ECs under different temperature treatments for 24 h.

**Figure 5 ijms-23-14047-f005:**
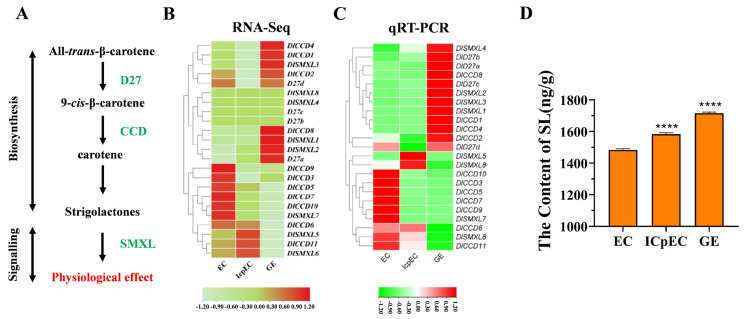
(**A**) Roles of CCD, D27, and SMXL genes in SLs pathway. (**B**) FPKM values of members of DlCCD, DlSMXL, and DlD27 family in longan SE. (**C**) Relative expression levels of DlCCD, DlSMXL, and DlD27 family in longan SE. (**D**) Changes in the endogenous SL content in longan SE. “****” (*p* < 0.001).

**Figure 6 ijms-23-14047-f006:**
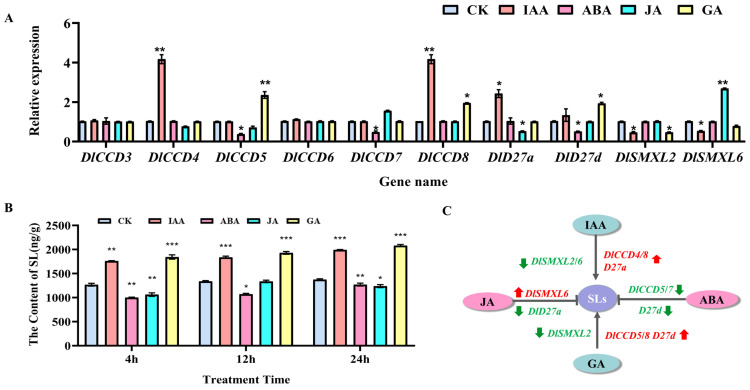
(**A**) The gene expression analysis and SL content changes of longan ECs treated with different hormones for 24 h. (**B**) Changes in the endogenous SL content of longan ECs under hormones treatment. (**C**) Schematic diagram of changes in the content of SLs and expression analysis of SL pathway genes respond to four hormone treatments for 24 h by qRT-PCR in longan ECs. Red and green mark indicate upregulated and downregulated genes. The arrow indicates promoting the process. The “T” line indicates inhibiting the process. Significant differences are indicated with “*” (*p* < 0.05), “**” (*p* < 0.01), “***” (*p* < 0.005).

**Figure 7 ijms-23-14047-f007:**
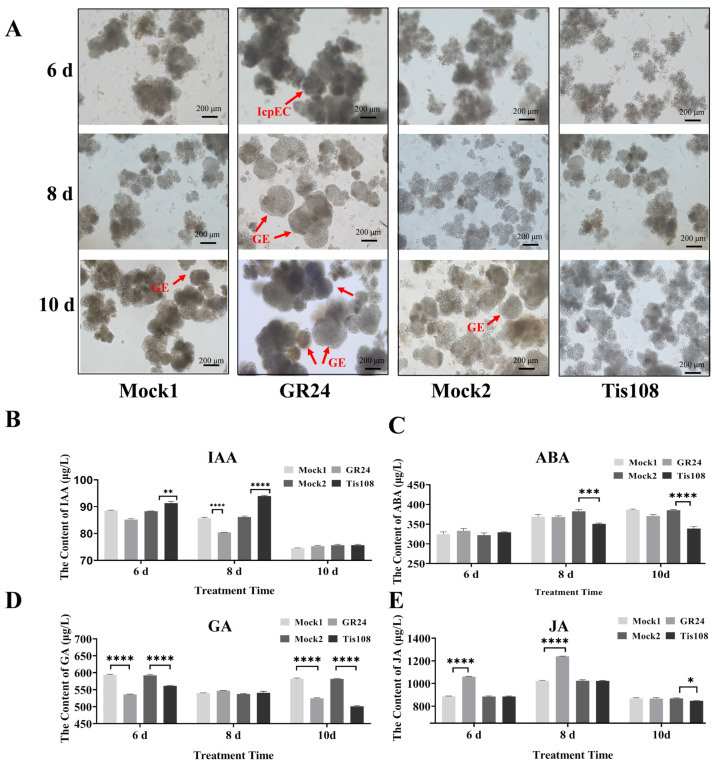
(**A**) Morphological change in embryogenic callus under GR24 and Tis108 treatment. The appearance globular embryo is marked by red arrows. (**B**–**E**) Changes in the endogenous IAA, ABA, GA, and JA content of longan somatic embryos upon GR24 and Tis108 treatment. Significant differences are indicated with “*” (*p* < 0.05), “**” (*p* < 0.01), “***” (*p* < 0.005), or “****” (*p* < 0.001).

**Figure 8 ijms-23-14047-f008:**
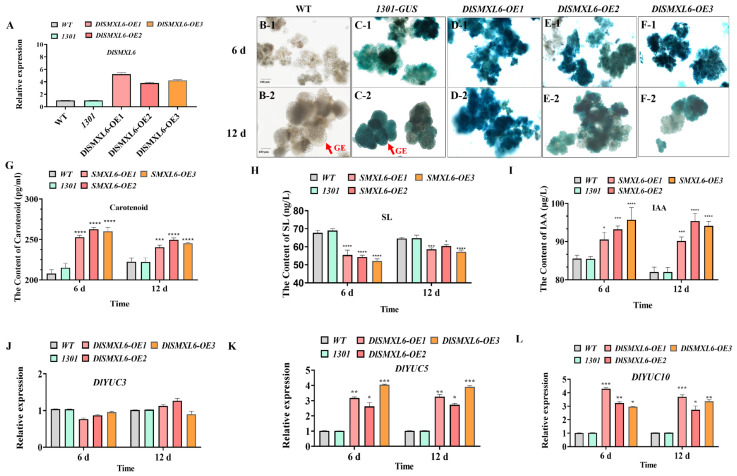
Morphological characteristics and physiological indexes determination of *DlSMXL6* overexpressing cell lines. (**A**) The expression levels of *DlSMXL6* in longan transgenic cell lines. (**B**–**F**) Phenotypical comparisons of wild-type (WT), transgenic *1301*-*GUS*, and *DlSMXL6-OEs*. 1301: expression vector; bar = 100 µm; the recombinant vector constructed of *1301*-*DlSMXL6*: *GUS*; (**G**–**I**) carotenoid, SLs, and IAA content changes of wild-type (WT), transgenic *1301-GUS* and *DlSMXL6-OEs*; (**J**–**L**) the expression levels of *DlYUC3*, *DlYUC5* and *DlYUC10* in longan transgenic cell lines. Significant differences are indicated with “*” (*p* < 0.05), “**” (*p* < 0.01), “***” (*p* < 0.005), or “****” (*p* < 0.001).

**Figure 9 ijms-23-14047-f009:**
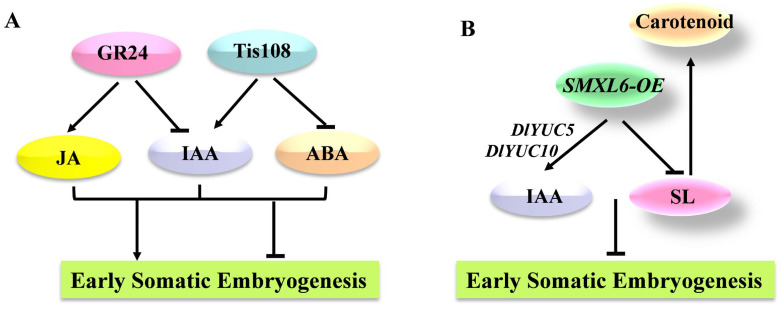
(**A**) The regulation network among GR24 and Tis108 with endogenous hormones during longan somatic embryo development. (**B**) The regulatory network composed of *DlSMXL6* during longan somatic embryo development. The arrow indicates promoting the process. The “T” line indicates inhibiting the process.

## Data Availability

The original contributions presented in the study are included in the article Material, further inquiries can be directed to the corresponding authors.

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
