# Peer review of "Whole Genome Analysis of SLs Pathway Genes and Functional Characterization of DlSMXL6 in Longan Early Somatic Embryo Development"

_ijms, 2022, doi:10.3390/ijms232214047_

Round 1

Reviewer 1 Report

The manuscript entitled “Whole Genome Analysis of SLs Pathway Genes and Functional Characterization of DlSMXL6 in Longan Early Somatic Embryo Development concerns the role of strigolactones in Dimocarpus longan regulation of early somatic embryogenesis. The authors identified genes involved in Sls signalling and biosynthesis in longan’s genome. They analyzed the expression profile of these genes in response to various treatment. The presented results indicate the cross-talk relation between Sls and other phytohormones, which is in fact known in literature for other than longan plant species.

This work can be valuable if the authors clearly state the purpose of the research and what it adds, what it concludes. I recommend the minor revision.

General comments:

·         In abstract: please do not replicate the view that sl are a newly discovered class of phytohormones. Research on these compounds has been going on long enough that they are unlikely to be referred to as new; the use of abbreviation of phytohormones’ names in abstract is not appropriate, difficult to read and understand. Please use the full names.

·         Introduction: the abbreviations like EC, GE should be explained at first use. Please avoid using too many abbreviations; it makes the work difficult to read when you have to keep going back to developments if you do not know them well; In the last paragraph of the introduction, the authors should clearly describe what the purpose of the research is. The reader does not actually find this out directly.

·         MM: what was the concentration of SLs  and Tis in the MS medium???

o   Was RNA digested with DNase, this information is missing?

o   Was the efficiency of qPCR reaction verified for each primer pair? The is no such information. The melting curves are also missing.

o   “via the 2− ΔΔCt method (Xu et al., 2021) – this is definitely wrong citation

o   “Agrobacterium tumefaciens strain (GV3101) by freeze-thaw method.” – please add  the citation to this standard method

o   Where is information concerning which statistical test was used in according to each experimental result??

·         The results description is chaotic, in my opinion for e.g. adding some schemes may be helpful to understand

·         Figures are poorly described. The reader has to guess what is on the figure, especially when it comes to photos. Lack of statistic description.

·         Discussion: The authors draw too premature conclusions with regard to the results obtained, eg. “These results showed that the functions of different DlCCD, DlD27 and DlSMXL members varied in different organs.” This is probably true but the presented results do not show it!

Author Response

Response to Reviewer 1 Comments

Point 1: In abstract: please do not replicate the view that sl are a newly discovered class of phytohormones. Research on these compounds has been going on long enough that they are unlikely to be referred to as new; the use of abbreviation of phytohormones’ names in abstract is not appropriate, difficult to read and understand. Please use the full names.

Response 1: Thank you for your valuable comment. We have modified the inappropriate view of sl are a newly discovered class of phytohormones and completed the full names in abstract.

Point 2: Introduction: the abbreviations like EC, GE should be explained at first use. Please avoid using too many abbreviations; it makes the work difficult to read when you have to keep going back to developments if you do not know them well; In the last paragraph of the introduction, the authors should clearly describe what the purpose of the research is. The reader does not actually find this out directly.

Response 2: We apologize for the unclear content about this section. We have completed the full names and the purpose of the research clearly in the MS.

Point 3: MM: what was the concentration of SLs and Tis in the MS medium???

Response 3: We apologize for the unclear content about this section.  GR24 (1 μM stock in acetone) and Tis108 (3 μM stock in DMSO) were added to MS liquid medium.

Point 4: Was RNA digested with DNase, this information is missing?

Response 4: We apologize for the misleading content about this section. We have added this part.

Point 5: Was the efficiency of qPCR reaction verified for each primer pair? The is no such information. The melting curves are also missing.

Response 5: Thank you for your valuable comment. We have verified the efficiency of qPCR reaction for each primer pair using the cDNA of longan SE, The melting curves were provided in the Figure S2.

Point 6: via the 2− ΔΔCt method (Xu et al., 2021) – this is definitely wrong citation

Response 2: Thank you for your valuable comment. We have modified the citation in the MS.

Point 7: “Agrobacterium tumefaciens strain (GV3101) by freeze-thaw method.” – please add the citation to this standard method

Response 7: We have add the method information in the MS.

Point 8: Where is information concerning which statistical test was used in according to each experimental result??

Response 8: Student’s t test was used in according to each experimental result. We have add the information in the MS.

Point 9: The results description is chaotic, in my opinion for e.g. adding some schemes may be helpful to understand.

Response 9: Thank you for your valuable comment. We have rewritten the results description and add the scheme in Figure 6.

Point 10: Figures are poorly described. The reader has to guess what is on the figure, especially when it comes to photos. Lack of statistic description.

Response 9: Thank you for your valuable comment. We have redrawn the Figures and rewritten the results description to better understanding.

Point 10: Discussion: The authors draw too premature conclusions with regard to the results obtained, eg. “These results showed that the functions of different DlCCD, DlD27 and DlSMXL members varied in different organs.” This is probably true but the presented results do not show it!

Response 10: Thank you for your valuable comment. We have carefully revised the discussion and modified the relevant sentences in the MS.

Reviewer 2 Report

The research results presented certainly have value and will find an audience. However, the manuscript as presented should not be published and needs thorough revision. First of all, the descriptions of the results do not match the graphs, graphics and tables presented. In addition, the graphics are of poor quality and it is difficult to read anything from them, and some of them are cut off. It is not clear whether the data for the analyses presented in the manuscript are new or previously published and taken from databases. Citation of the figures is not correct. It would also be good to arrange the graphs in the order of description in the text. The authors should also work on the language to make the descriptions more understandable. I have included some of the more detailed comments below.

“To investigate the gene expression patterns and endogenous SLs content changes in response to different hormone treatments, we selected ten SLs biosynthesis and responding genes for qRT-PCR analysis and determined the content of SLs.” – this sentence does not fit to the paragraph 2.1.

“The signal peptide and genomic location were analyzed with SignalP4.0” – what do you mean by ‘genomic location’? How genomic location can be analyzed with SignalP tool?

"EC, IcpEC and GE". - what is the meaning of these abbreviations? Nowhere in the manuscript are they explained

FPKM expression profile analysis of genes involved in the SL pathway in Longan was extracted from four transcriptome datasets - Are these available in public databases? Is the embryogenic culture dataset (EC, IcpEC and GE) already published? Did the authors do the sequencing AND analysis of the data or did they use available results, it's not entirely clear to me. "PacBio Sequel was used to construct third-generation genomic data of high quality HiC sequencing for longan." - this sentence indicates that sequencing was performed as part of the research described

“The analysis of FPKM expression profile of genes involved in SL pathway in Longan were extracted from four transcriptome datasets” – Are these available in public databases? Is the embryogenic culture dataset (EC, IcpEC and GE) already published? Did the authors do the sequencing AND analysis of the data or did they use available results, it's not entirely clear to me. „PacBio Sequel was used to construct the third-generation genomic high-quality HiC sequencing data for longan” - this sentence indicates that sequencing was performed as part of the described research

“Analysis of the conserved motif showed that DlD27d had 13 motifs with the highest numbers, followed by DlD27a with 11 motif.” – The results decribed in this paragraph do not correspond to those shown in Fig S1.

It would be a good idea to rearrange Figure S1, because it is not clear what is part A or B. It is not clear which gene is which in part C. I can only assume that the order is the same as in part A, and the legend for C is misplaced.

“Figure 5. A Roles of CCD, D27 and SMXL genes in SLs pathway.” - It would have been good if the authors had explained the biosynthesis process of stringolactones a bit more in the introduction. The authors did not mention a word about the D27 group and its involvement in the biosynthesis of SLs

DlCCD10 was predicted to localize in the cytoplasm. DlCCD6 and DlCCD7 were found in peroxisome. DlSMXL6 was predicted to localize in the extracellular matrix. Six members were found primarily in cytoplasm, including DlS-MXL2, DlSMXL3, DlSMXL7, DlD27a, DlD27c and DlD27d.” – The decribed results do not correspond to those shown in Table S2

Paragraph beggining with “As shown in Figure 1A, 23 putative DlCCD, DlSMXL and DlD27 family members were randomly and unevenly distributed on ten chromosomes of longan…” describes the results differently than in the figures and tables presented and cited.

“We firstly determined the endogenous SL content of longan ECs under different hormone treatments, the results showed that exogenous IAA and GA significantly increased the SLs content compared to control, while ABA and JA decreased the SLs content in longan EC (Figure 6C).” – there is no control showed on the figure 6c

“The results showed that the expression of DlCCD4, DlCCD6, DlCCD8, DlD27a and DlD27d were up-regulated, while DlSMXL2 and DlSMXL6 were downregulated by IAA treatment” – but most of the results are insignificant. It is difficult to read the value of the expression level and compare whether the level is the same or different. The authors should consider adding a table with numbers to the Supplements.

“Compared with WT and 1301-GUS, DlYUC5 and DlYUC10 were obviously up-regulated and DlYUC5 was down-regulated in the DlSMXL6 overexpression cell lines, which may play important roles in longan SE” – So the expression of the DlYUC5 is obviously up- or down-regulated?

The discussion should be more developed, the results with which the authors compare theirs should be better described. “The physiological interactions between SL and GA have been reported previously (Toh et al., 2012; Saint et al., 2013; Ni et al., 2015; Ni et al., 2017).” –. As it stands, the reader would have to read all the cited publications to find out what specifically the authors of this manuscript are referring to

Author Response

Response to Reviewer 2 Comments

Point 1:“To investigate the gene expression patterns and endogenous SLs content changes in response to different hormone treatments, we selected ten SLs biosynthesis and responding genes for qRT-PCR analysis and determined the content of SLs.” – this sentence does not fit to the paragraph 2.1.

Response 1: Thank you for your valuable comment. We have deleted the sentence.

Point 2: 2)“The signal peptide and genomic location were analyzed with SignalP4.0” – what do you mean by ‘genomic location’? How genomic location can be analyzed with SignalP tool?

Response 2: We apologize for the misleading content about this section, we have modified the sentence.

Point 3: "EC, IcpEC and GE". - what is the meaning of these abbreviations? Nowhere in the manuscript are they explained.

Response 3: Thank you for your valuable comment. ECs (embryogenic callus), IcpEC (Incomplete compact proembryogenic culture) and GEs (globular embryos). We have completed the full names in the manuscript.

Point 4: FPKM expression profile analysis of genes involved in the SL pathway in Longan was extracted from transcriptome datasets - Are these available in public databases? Is the embryogenic culture dataset (EC, IcpEC and GE) already published? Did the authors do the sequencing AND analysis of the data or did they use available results, it's not entirely clear to me. "PacBio Sequel was used to construct third-generation genomic data of high quality HiC sequencing for longan." - this sentence indicates that sequencing was performed as part of the research described

Response 4: 1) We have completed the transcriptome datasets information in the manuscript. The reviewers can log on the NCBI to review (ORCID iD: 0000-0002-2911-9780, password: buliang84).

  • We have modified this sentence “PacBio Sequel was used to construct the third-generation genomic high-quality HiC sequencing data for longan” in the manuscript.
  • Point 5:“Analysis of the conserved motif showed that DlD27d had 13 motifs with the highest numbers, followed by DlD27a with 11 motif.”-The results decribed in this paragraph do not correspond to those shown in Fig S1.

Response 5: Thank you for your valuable comment. We apologize for the mistaken content about this section, we have modified the sentence.

Point 6: It would be a good idea to rearrange Figure S1, because it is not clear what is part A or B. It is not clear which gene is which in part C. I can only assume that the order is the same as in part A, and the legend for C is misplaced.

Response 6: Thank you for your valuable comment. We have rearranged Figure S1.

Point 7: “Figure 5. A Roles of CCD, D27 and SMXL genes in SLs pathway.” - It would have been good if the authors had explained the biosynthesis process of stringolactones a bit more in the introduction. The authors did not mention a word about the D27 group and its involvement in the biosynthesis of SLs

Response 7: Thank you for your valuable comment. We have added the content about the D27 group and its involvement in the biosynthesis of SLs in the introduction.

Point 8: “DlCCD10 was predicted to localize in the cytoplasm. DlCCD6 and DlCCD7 were found in peroxisome. DlSMXL6 was predicted to localize in the extracellular matrix. Six members were found primarily in cytoplasm, including DlS-MXL2, DlSMXL3, DlSMXL7, DlD27a, DlD27c and DlD27d.” – The decribed results do not correspond to those shown in Table S2

Response 8: Thank you for your valuable comment. We apologize for the mistaken content about this section, we have revised the content.

Point 9: Paragraph beggining with “As shown in Figure 1A, 23 putative DlCCD, DlSMXL and DlD27 family members were randomly and unevenly distributed on ten chromosomes of longan…” describes the results differently than in the figures and tables presented and cited.

Response 9: Thank you for your valuable comment. We apologize for the mistaken content about this section, we have modified the content.

Point 10: “We firstly determined the endogenous SL content of longan ECs under different hormone treatments, the results showed that exogenous IAA and GA significantly increased the SLs content compared to control, while ABA and JA decreased the SLs content in longan EC (Figure 6C).” – there is no control showed on the figure 6c

Response 9: Thank you for your valuable comment. We have redraw the figure 6 to better understanding.

Point 10:The results showed that the expression of DlCCD4, DlCCD6, DlCCD8, DlD27a and DlD27d were up-regulated, while DlSMXL2 and DlSMXL6 were downregulated by IAA treatment” – but most of the results are insignificant. It is difficult to read the value of the expression level and compare whether the level is the same or different. The authors should consider adding a table with numbers to the Supplements.

Response 10: Thank you for your valuable comment. We have added a table with numbers to the Supplements (Table S3).

Point 11:“Compared with WT and 1301-GUS, DlYUC5 and DlYUC10 were obviously up-regulated and DlYUC5 was down-regulated in the DlSMXL6 overexpression cell lines, which may play important roles in longan SE” – So the expression of the DlYUC5 is obviously up- or down-regulated?

Response 11: Thank you for your valuable comment. Thank you for your valuable comment. We apologize for the mistaken, the expression of the DlYUC5 is obviously up-regulated.

Point 11: The discussion should be more developed, the results with which the authors compare theirs should be better described. “The physiological interactions between SL and GA have been reported previously (Toh et al., 2012; Saint et al., 2013; Ni et al., 2015; Ni et al., 2017).” –. As it stands, the reader would have to read all the cited publications to find out what specifically the authors of this manuscript are referring to.

Response 11: Thank you for your valuable comment. We have rewritten the discussion section and completed the content of cited publications.

Round 2

Reviewer 2 Report

Review 2

1)                The results described in the text do not correspond to those shown in Fig S1.

“DlSMXL5 and DlSMXL6 only contained motif 16 and motif 13, respectively. Motif 1, motif 7, motif 5, motif 16, motif 2, motif 15, and motif 12 were present in all family members, indicating a high conservation in longan evolution”

2)                Please check the citation method eg. “(Waters, Brewer, Bussell, Smith, & Beveridge, 2012)”

3)                Delete the paragraph: “As shown in Figure 1A, 23 putative DlCCD, DlSMXL and DlD27 family members were randomly and unevenly distributed on ten chromosomes of longan. The highest number of DlCCD, DlSMXL and DlD27 family genes was observed in chromosome1 (Chr1) with 5 members. DlSMXL family genes were located on Chr1, Chr5, Chr7, Chr12 and Chr14. DlD27 genes were located on Chr3, Chr4 and Chr10. Chr4, Chr10 and Chr12 only contained one number.” It is added twice.

4)                Sentence “CCD7 were highly expressed in the tomato fruit (Vogel et al., 2010).” Is also added two times

5)                I am not a native speaker, but I believe the authors should carry out a linguistic correction especially of the newly added passages.

Author Response

Dear editor and reviewers

Thank you and reviewers very much for constructive suggestions and comments, which greatly improve our manuscript. All of the changes to the manuscript are indicated in the text using track changes. We wish that our answers could be as complete as possible to the question of reviewers.

Point 1:“The results described in the text do not correspond to those shown in Fig S1.

DlSMXL5 and DlSMXL6 only contained motif 16 and motif 13, respectively. Motif 1, motif 7, motif 5, motif 16, motif 2, motif 15, and motif 12 were present in all family members, indicating a high conservation in longan evolution”

Response 1: Thank you for your valuable comment. We have modified the sentence in the MS.

Point 2:  Please check the citation method eg. “(Waters, Brewer, Bussell, Smith, & Beveridge, 2012)”

Response 2: Thank you for your valuable comment. We have checked the citation method and modified it in the MS.

Point 3: Delete the paragraph: “As shown in Figure 1A, 23 putative DlCCD, DlSMXL and DlD27 family members were randomly and unevenly distributed on ten chromosomes of longan. The highest number of DlCCD, DlSMXL and DlD27 family genes was observed in chromosome1 (Chr1) with 5 members. DlSMXL family genes were located on Chr1, Chr5, Chr7, Chr12 and Chr14. DlD27 genes were located on Chr3, Chr4 and Chr10. Chr4, Chr10 and Chr12 only contained one number.” It is added twice.

Response 3: Thank you for your valuable comment. We have deleted the paragraph in the manuscript.

Point 4:  Sentence “CCD7 were highly expressed in the tomato fruit (Vogel et al., 2010).” Is also added two times.

Response 4: Thank you for your valuable comment. We have deleted the sentence in the manuscript.

Point 5:   I am not a native speaker, but I believe the authors should carry out a linguistic correction especially of the newly added passages.

Response 5: Thank you for your valuable comment. We have carefully revised the manuscript according to the reviewers' comments, and improve the MS English by English-speaking experts in related fields.